# Next generation reservoir computing

Daniel J. Gauthier [1,2✉], Erik Bollt[3,4], Aaron Griffith [1] & Wendson A. S. Barbosa [1]

Reservoir computing is a best-in-class machine learning algorithm for processing information generated by dynamical systems using observed time-series data. Importantly, it requires very small training data sets, uses linear optimization, and thus requires minimal computing resources. However, the algorithm uses randomly sampled matrices to define the underlying recurrent neural network and has a multitude of metaparameters that must be optimized. Recent results demonstrate the equivalence of reservoir computing to nonlinear vector autoregression, which requires no random matrices, fewer metaparameters, and provides interpretable results. Here, we demonstrate that nonlinear vector autoregression excels at reservoir computing benchmark tasks and requires even shorter training data sets and training time, heralding the next generation of reservoir computing.

[1] The Ohio State University, Department of Physics, 191 West Woodruff Ave., Columbus, OH 43210, USA. [2] ResCon Technologies, LLC, PO Box 21229 Columbus, OH 43221, USA. [3] Clarkson University, Department of Electrical and Computer Engineering, Potsdam, NY 13669, USA. [4] Clarkson Center for Complex Systems Science (C3S2), Potsdam, NY 13699, USA. ✉email: gauthier.51@osu.edu

A dynamical system evolves in time, with examples including the Earth's weather system and human-built devices such as unmanned aerial vehicles. One practical goal is to develop models for forecasting their behavior. Recent machine learning (ML) approaches can generate a model using only observed data, but many of these algorithms tend to be data hungry, requiring long observation times and substantial computational resources.

Reservoir computing[1,2] is an ML paradigm that is especially well-suited for learning dynamical systems. Even when systems display chaotic[3] or complex spatiotemporal behaviors[4], which are considered the hardest-of-the-hard problems, an optimized reservoir computer (RC) can handle them with ease.

As described in greater detail in the next section, an RC is based on a recurrent artificial neural network with a pool of interconnected neurons—the reservoir, an input layer feeding observed data $\mathbf{X}$ to the network, and an output layer weighting the network states as shown in Fig. 1. To avoid the vanishing gradient problem[5] during training, the RC paradigm randomly assigns the input-layer and reservoir link weights. Only the weights of the output links $\mathbf{W}_{out}$ are trained via a regularized linear least-squares optimization procedure[6]. Importantly, the regularization parameter α is set to prevent overfitting to the training data in a controlled and well understood manner and makes the procedure noise tolerant. RCs perform as well as other ML methods, such as Deep Learning, on dynamical systems tasks but have substantially smaller data set requirements and faster training times[7,8].

Using random matrices in an RC presents problems: many perform well, but others do not all and there is little guidance to select good or bad matrices. Furthermore, there are several RC metaparameters that can greatly affect its performance and require optimization[9–13]. Recent work suggests that good matrices and metaparameters can be identified by determining whether the reservoir dynamics $r$ synchronizes in a generalized sense to $\mathbf{X}$[14,15], but there are no known design rules for obtaining generalized synchronization.

Recent RC research has identified requirements for realizing a general, universal approximator of dynamical systems. A universal approximator can be realized using an RC with nonlinear activation at nodes in the recurrent network and an output layer (known as the feature vector) that is a weighted linear sum of the network nodes under the weak assumptions that the dynamical system has bounded orbits[16].

Less appreciated is the fact that an RC with linear activation nodes combined with a feature vector that is a weighted sum of nonlinear functions of the reservoir node values is an equivalently powerful universal approximator[16,17]. Furthermore, such an RC is mathematically identical to a nonlinear vector autoregression (NVAR) machine[18]. Here, no reservoir is required: the feature vector of the NVAR consists of $k$ time-delay observations of the dynamical system to be learned and nonlinear functions of these observations, as illustrated in Fig. 1, a surprising result given the apparent lack of a reservoir!

These results are in the form of an existence proof: There exists an NVAR that can perform equally well as an optimized RC and, in turn, the RC is implicit in an NVAR. Here, we demonstrate that it is easy to design a well-performing NVAR for three challenging RC benchmark problems: (1) forecasting the short-term dynamics; (2) reproducing the long-term climate of a

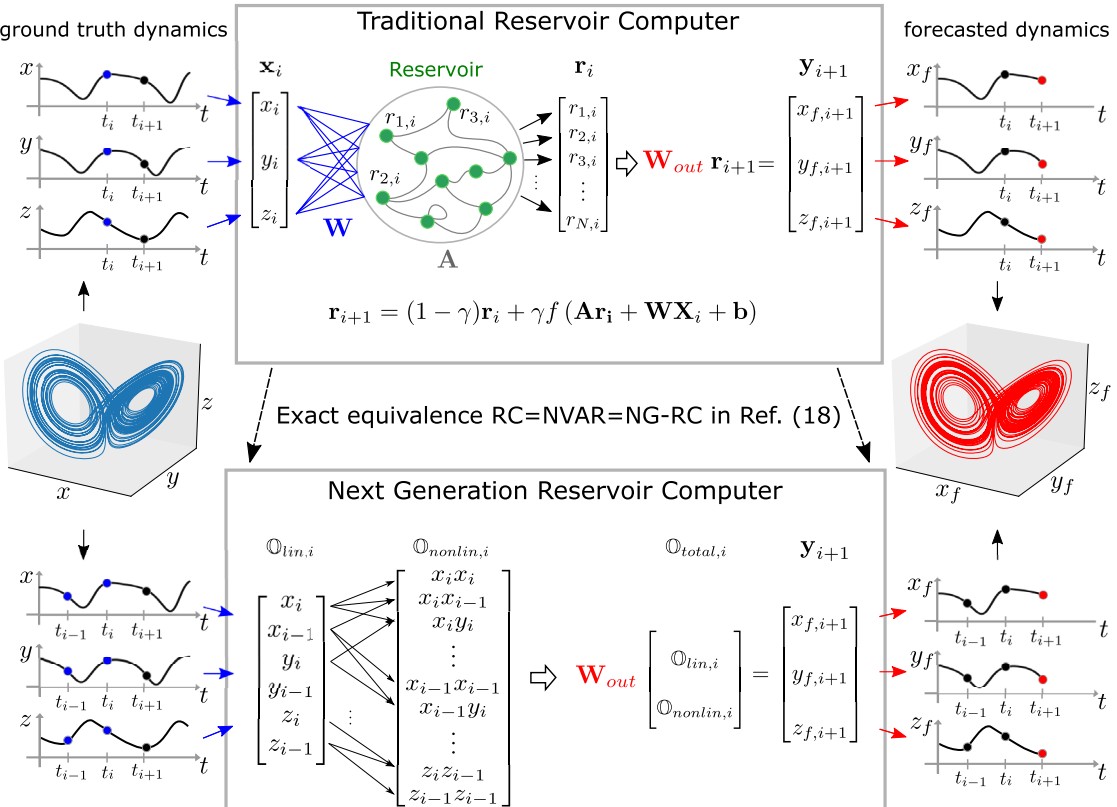

**Fig. 1 A traditional RC is implicit in an NG-RC.** (top) A traditional RC processes time-series data associated with a strange attractor (blue, middle left) using an artificial recurrent neural network. The forecasted strange attractor (red, middle right) is a linear weight of the reservoir states. (bottom) The NG-RC performs a forecast using a linear weight of time-delay states (two times shown here) of the time series data and nonlinear functionals of this data (quadratic functional shown here).

chaotic system (that is, reconstructing the attractors shown in Fig. 1); and (3) inferring the behavior of unseen data of a dynamical system.

Predominantly, the recent literature has focused on the first benchmark of short-term forecasting of stochastic processes time-series data[16], but the importance of high-accuracy forecasting and inference of unseen data cannot be overstated. The NVAR, which we call the next generation RC (NG-RC), displays state-of-the-art performance on these tasks because it is associated with an implicit RC, and uses exceedingly small data sets and side-steps the random and parametric difficulties of directly implementing a traditional RC.

We briefly review traditional RCs and introduce an RC with linear reservoir nodes and a nonlinear output layer. We then introduce the NG-RC and discuss the remaining metaparameters, introduce two model systems we use to showcase the performance of the NG-RC, and present our findings. Finally, we discuss the implications of our work and future directions.

The purpose of an RC illustrated in the top panel of Fig. 1 is to broadcast input data $\mathbf{X}$ into the higher-dimensional reservoir network composed of $N$ interconnected nodes and then to combine the resulting reservoir state into an output $\mathbf{Y}$ that closely matches the desired output $\mathbf{Y}_d$. The strength of the node-to-node connections, represented by the connectivity (or adjacency) matrix $\mathbf{A}$, are chosen randomly and kept fixed. The data to be processed $\mathbf{X}$ is broadcast into the reservoir through the input layer with fixed random coefficients $\mathbf{W}$. The reservoir is a dynamic system whose dynamics can be represented by

$$\mathbf{r}_{i+1} = (1 - \gamma)\mathbf{r}_i + \gamma f(\mathbf{A}\mathbf{r}_i + \mathbf{W}\mathbf{X}_i + \mathbf{b}), \qquad (1)$$

where $\mathbf{r}_i = [r_{1,i}, r_{2,i}, ..., r_{N,i}]^T$ is an $N$-dimensional vector with component $r_{j,i}$ representing the state of the $j$th node at the time $t_i$, $\gamma$ is the decay rate of the nodes, $f$ an activation function applied to each vector component, and $\mathbf{b}$ is a node bias vector. For simplicity, we choose $\gamma$ and $\mathbf{b}$ the same for all nodes. Here, time is discretized at a finite sample time $dt$ and $i$ indicates the $i$th time step so that $dt = t_{i+1} - t_i$. Thus, the notations $\mathbf{r}_i$ and $\mathbf{r}_{i+1}$ represent the reservoir state in consecutive time steps. The reservoir can also equally well be represented by continuous-time ordinary differential equations that may include the possibility of delays along the network links[19].

The output layer expresses the RC output $\mathbf{Y}_{i+1}$ as a linear transformation of a feature vector $\mathbb{O}_{\text{total},i+1}$, constructed from the reservoir state $\mathbf{r}_{i+1}$, through the relation

$$\mathbf{Y}_{i+1} = \mathbf{W}_{\text{out}}\mathbb{O}_{\text{total},i+1}, \qquad (2)$$

where $\mathbf{W}_{\text{out}}$ is the output weight matrix and the subscript total indicates that it can be composed of constant, linear, and non-linear terms as explained below. The standard approach, commonly used in the RC community, is to choose a nonlinear activation function such as $f(x) = \tanh(x)$ for the nodes and a linear feature vector $\mathbb{O}_{\text{total},i+1} = \mathbb{O}_{\text{lin},i+1} = \mathbf{r}_{i+1}$ in the output layer. The RC is trained using supervised training via regularized least-squares regression. Here, the training data points generate a block of data contained in $\mathbb{O}_{\text{total}}$ and we match $\mathbf{Y}$ to the desired output $\mathbf{Y}_d$ in a least-square sense using Tikhonov regularization so that $\mathbf{W}_{\text{out}}$ is given by

$$\mathbf{W}_{\text{out}} = \mathbf{Y}_d\mathbb{O}_{\text{total}}{}^T(\mathbb{O}_{\text{total}}\mathbb{O}_{\text{total}}{}^T + \alpha\mathbf{I})^{-1}, \qquad (3)$$

where the regularization parameter $\alpha$, also known as ridge parameter, is set to prevent overfitting to the training data and $\mathbf{I}$ is the identity matrix.

A different approach to RC is to move the nonlinearity from the reservoir to the output layer[16,18]. In this case, the reservoir nodes are chosen to have a linear activation function $f(\mathbf{r}) = \mathbf{r}$, while the

feature vector $\mathbb{O}_{\text{total}}$ becomes nonlinear. A simple example of such RC is to extend the standard linear feature vector to include the squared values of all nodes, which are obtained through the Hadamard product $\mathbf{r} \odot \mathbf{r} = [r_1^2, r_2^2, \ldots, r_N^2]^T$[18]. Thus, the nonlinear feature vector is given by

$$\mathbb{O}_{\text{total}} = \mathbf{r} \oplus (\mathbf{r} \odot \mathbf{r}) = [r_1, r_2, \ldots, r_N, r_1^2, r_2^2, \ldots, r_N^2]^T, \qquad (4)$$

where $\oplus$ represents the vector concatenation operation. A linear reservoir with a nonlinear output is an equivalently powerful universal approximator[16] and shows comparable performance to the standard RC[18].

In contrast, the NG-RC creates a feature vector directly from the discretely sample input data with no need for a neural network. Here, $\mathbb{O}_{\text{total}} = c \oplus \mathbb{O}_{\text{lin}} \oplus \mathbb{O}_{\text{nonlin}}$, where $c$ is a constant and $\mathbb{O}_{\text{nonlin}}$ is a nonlinear part of the feature vector. Like a traditional RC, the output is obtained using these features in Eq. 3. We now discuss forming these features.

The linear features $\mathbb{O}_{\text{lin},i}$ at time step $i$ is composed of observations of the input vector $\mathbf{X}$ at the current and at $k-1$ previous times steps spaced by $s$, where $(s-1)$ is the number of skipped steps between consecutive observations. If $\mathbf{X}_i = [x_{1,i}, x_{2,i}, \ldots, x_{d,i}]^T$ is a $d$-dimensional vector, $\mathbb{O}_{\text{lin},i}$ has $d\,k$ components, and is given by

$$\mathbb{O}_{\text{lin},i} = \mathbf{X}_i \oplus \mathbf{X}_{i-s} \oplus \mathbf{X}_{i-2s} \oplus \ldots \oplus \mathbf{X}_{i-(k-1)s}. \qquad (5)$$

Based on the general theory of universal approximators[16,20], $k$ should be taken to be infinitely large. However, it is found in practice that the Volterra series converges rapidly, and hence truncating $k$ to small values does not incur large error. This can also be motivated by considering numerical integration methods of ordinary differential equations where only a few subintervals (steps) in a multistep integrator are needed to obtain high accuracy. We do not subdivide the step size here, but this analogy motivates why small values of $k$ might give good performance in the forecasting tasks considered below.

An important aspect of the NG-RC is that its warm-up period only contains $(sk)$ time steps, which are needed to create the feature vector for the first point to be processed. This is a dramatically shorter warm-up period in comparison to traditional RCs, where longer warm-up times are needed to ensure that the reservoir state does not depend on the RC initial conditions. For example, with $s = 1$ and $k = 2$ as used for some examples below, only two warm-up data points are needed. A typical warm-up time in traditional RC for the same task can be upwards of $10^3$ to $10^5$ data points[12,14]. A reduced warm-up time is especially important in situations where it is difficult to obtain data or collecting additional data is too time-consuming.

For the case of a driven dynamical system, $\mathbb{O}_{\text{lin}}(t)$ also includes the drive signal[21]. Similarly, a system in which one or more accessible system parameters are adjusted, $\mathbb{O}_{\text{lin}}(t)$ also includes these parameters[21,22].

The nonlinear part $\mathbb{O}_{\text{nonlin}}$ of the feature vector is a nonlinear function of $\mathbb{O}_{\text{lin}}$. While there is great flexibility in choosing the nonlinear functionals, we find that polynomials provide good prediction ability. Polynomial functionals are the basis of a Volterra representation for dynamical systems[20] and hence they are a natural starting point. We find that low-order polynomials are enough to obtain high performance.

All monomials of the quadratic polynomial, for example, are captured by the outer product $\mathbb{O}_{\text{lin}} \otimes \mathbb{O}_{\text{lin}}$, which is a symmetric matrix with $(dk)^2$ elements. A quadratic nonlinear feature vector $\mathbb{O}_{\text{nonlinear}}^{(2)}$, for example, is composed of the $(dk)(dk+1)/2$ unique monomials of $\mathbb{O}_{\text{lin}} \otimes \mathbb{O}_{\text{lin}}$, which are given by the upper triangular elements of the outer product tensor. We define $\lceil\otimes\rceil$ as the operator that collects the unique monomials in a vector. Using

this notation, a $p$-order polynomial feature vector is given by

$$\mathbb{O}^{(p)}_{\text{nonlinear}} = \mathbb{O}_{\text{lin}} \lceil \otimes \rceil \mathbb{O}_{\text{lin}} \lceil \otimes \rceil \ldots \lceil \otimes \rceil \mathbb{O}_{\text{lin}} \qquad (6)$$

with $\mathbb{O}_{\text{lin}}$ appearing $p$ times.

Recently, it was mathematically proven that the NVAR method is equivalent to a linear RC with polynomial nonlinear readout[18]. This means that every NVAR implicitly defines the connectivity matrix and other parameters of a traditional RC described above and that every linear polynomial-readout RC can be expressed as an NVAR. However, the traditional RC is more computationally expensive and requires optimizing many meta-parameters, while the NG-RC is more efficient and straightforward. The NG-RC is doing the same work as the equivalent traditional RC with a full recurrent neural network, but we do not need to find that network explicitly or do any of the costly computation associated with it.

We now introduce models and tasks we use for showcasing the performance of NG-RC. For one of the forecasting tasks and the inference task discussed in the next section, we generate training and testing data by numerically integrating a simplified model of a weather system[23] developed by Lorenz in 1963. It consists of a set of three coupled nonlinear differential equations given by

$$\dot{x} = 10(y - x), \quad \dot{y} = x(28 - z) - y, \quad \dot{z} = xy - 8z/3, \qquad (7)$$

where the state $\mathbf{X}(t) \equiv [x(t), y(t), z(t)]^T$ is a vector whose components are Rayleigh–Bénard convection observables. It displays deterministic chaos, sensitive dependence to initial conditions—the so-called butterfly effect—and the phase space trajectory forms a strange attractor shown in Fig. 1. For future reference, the Lyapunov time for Eq. 7, which characterizes the divergence timescale for a chaotic system, is 1.1-time units. Below, we refer to this system as Lorenz63.

We also explore using the NG-RC to predict the dynamics of a double-scroll electronic circuit[24] whose behavior is governed by

$$\dot{V}_1 = V_1/R_1 - \Delta V/R_2 - 2I_r \, \sinh(\beta \Delta V),$$
$$\dot{V}_2 = \Delta V/R_2 + 2I_r \, \sinh(\beta \Delta V) - I, \qquad (8)$$
$$\dot{I} = V_2 - R_4 I$$

in dimensionless form, where $\Delta V = V_1 - V_2$. Here, we use the parameters $R_1 = 1.2$, $R_2 = 3.44$, $R_4 = 0.193$, $\beta = 11.6$, and $I_r = 2.25 \times 10^{-5}$, which give a Lyapunov time of 7.81-time units.

We select this system because the vector field is not of a polynomial form and $\Delta V$ is large enough at some times that a truncated Taylor series expansion of the sinh function gives rise to large differences in the predicted attractor. This task demonstrates that the polynomial form of the feature vector can work for nonpolynomial vector fields as expected from the theory of Volterra representations of dynamical systems[20].

In the two forecasting tasks presented below, we use an NG-RC to forecast the dynamics of Lorenz63 and the double-scroll system using one-step-ahead prediction. We start with a listening phase, seeking a solution to $\mathbf{X}(t + dt) = \mathbf{W}_{\text{out}} \mathbb{O}_{\text{total}}(t)$, where $\mathbf{W}_{\text{out}}$ is found using Tikhonov regularization[6]. During the forecasting (testing) phase, the components of $\mathbf{X}(t)$ are no longer provided to the NG-RC and the predicted output is fed back to the input. Now, the NG-RC is an autonomous dynamical system that predicts the systems' dynamics if training is successful.

The total feature vector used for the Lorenz63 forecasting task is given by

$$\mathbb{O}_{\text{total}} = c \oplus \mathbb{O}_{\text{lin}} \oplus \mathbb{O}^{(2)}_{\text{nonlinear}}, \qquad (9)$$

which has $[1 + d\,k + (d\,k)\,(d\,k+1)/2]$ components.

For the double-scroll system forecasting task, we notice that the attractor has odd symmetry and has zero mean for all variables for the parameters we use. To respect these characteristics, we

take

$$\mathbb{O}_{\text{total}} = \mathbb{O}_{\text{lin}} \oplus \mathbb{O}^{(3)}_{\text{nonlinear}} \qquad (10)$$

which has $[d\,k + (d\,k)\,(d\,k+1)\,(d\,k+2)/6]$ components.

For these forecasting tasks, the NG-RC learns simultaneously the vector field and an efficient one-step-ahead integrator to find a mapping from one time to the next without having to learn each separately as in other nonlinear state estimation approaches[25–28]. The one-step-ahead mapping is known as the flow of the dynamical system and hence the NG-RC learns the flow. To allow the NG-RC to focus on the subtle details of this process, we use a simple Euler-like integration step as a lowest-order approximation to a forecasting step by modifying Eq. 2 so that the NG-RC learns the difference between the current and future step. To this end, Eq. 2 is replaced by

$$\mathbf{X}_{i+1} = \mathbf{X}_i + \mathbf{W}_{\text{out}} \mathbb{O}_{\text{total},i}. \qquad (11)$$

In the third task, we provide the NG-RC with all three Lorenz63 variables during training with the goal of inferring the next-step-ahead prediction of one of the variables from the others. During testing, we only provide it with the $x$ and $y$ variables and infer the $z$ variable. This task is important for applications where it is possible to obtain high-quality information about a dynamical variable in a laboratory setting, but not in field deployment. In the field, the observable sensory information is used to infer the missing data.

## Results

For the first task, the ground-truth Lorenz63 strange attractor is shown in Fig. 2a. The training phase uses only the data shown in Fig. 2b–d, which consists of 400 data points for each variable with $dt = 0.025$, $k = 2$, and $s = 1$. The training compute time is <10 ms using Python running on a single-core desktop processor (see Methods). Here, $\mathbb{O}_{\text{total}}$ has 28 components and $\mathbf{W}_{\text{out}}$ has dimension $(3 \times 28)$. The set needs to be long enough for the phase-space trajectory to explore both wings of the strange attractor. The plot is overlaid with the NG-RC predictions during training; no difference is visible on this scale.

The NG-RC is then placed in the prediction phase; a qualitative inspection of the predicted (Fig. 2e) and true (Fig. 2a) strange attractors shows that they are very similar, indicating that the NG-RC reproduces the long-term climate of Lorenz63 (benchmark problem 2). As seen in Fig. 2f–h, the NG-RC does a good job of predicting Lorenz63 (benchmark 1), comparable to an optimized traditional RC[3,12,14] with 100s to 1000s of reservoir nodes. The NG-RC forecasts well out to ~5 Lyapunov times.

In Supplementary Note 1, we give other quantitative measurements of the accuracy of the attractor reconstruction and the values of $\mathbf{W}_{\text{out}}$ in Supplementary Note 2; there are many components that have substantial weights and that do not appear in the vector field of Eq. 7, where the vector field is the right-hand-side of the differential equations. This gives quantitative information regarding the difference between the flow and the vector field.

Because the Lyapunov time for the double-scroll system is much longer than for the Lorenz63 system, we extend the training time of the NG-RC from 10 to 100 units to keep the number of Lyapunov times covered during training similar for both cases. To ensure a fair comparison to the Lorenz63 task, we set $dt = 0.25$. With these two changes and the use of the cubic monomials, as given in Eq. 10, with $d = 3$, $k = 2$, and $s = 1$ for a total of 62 features in $\mathbb{O}_{\text{total}}$, the NG-RC uses 400 data points for each variable during training, exactly as in the Lorenz63 task.

Other than these modifications, our method for using the NG-RC to forecast the dynamics of this system proceeds exactly as for

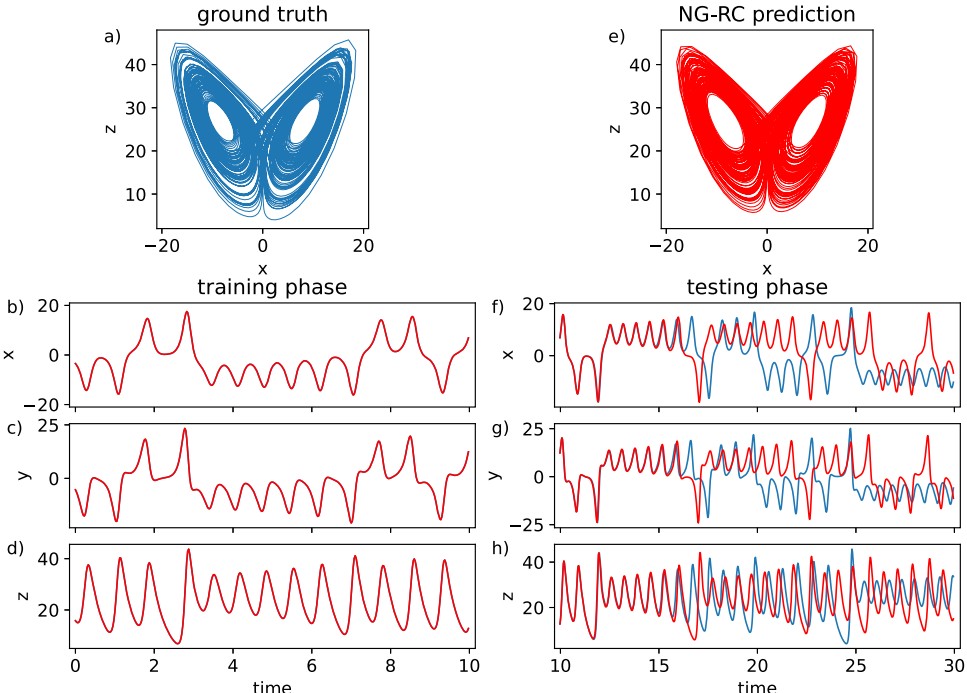

**Fig. 2 Forecasting a dynamical system using the NG-RC.** True (**a**) and predicted (**e**) Lorenz63 strange attractors. **b–d** Training data set with overlayed predicted behavior with $\alpha = 2.5 \times 10^{-6}$. The normalized root-mean-square error (NRMSE) over one Lyapunov time during the training phase is $1.06 \pm 0.01 \times 10^{-4}$, where the uncertainty is the standard error of the mean. **f–h** True (blue) and predicted datasets during the forecasting phase (NRMSE $= 2.40 \pm 0.53 \times 10^{-3}$).

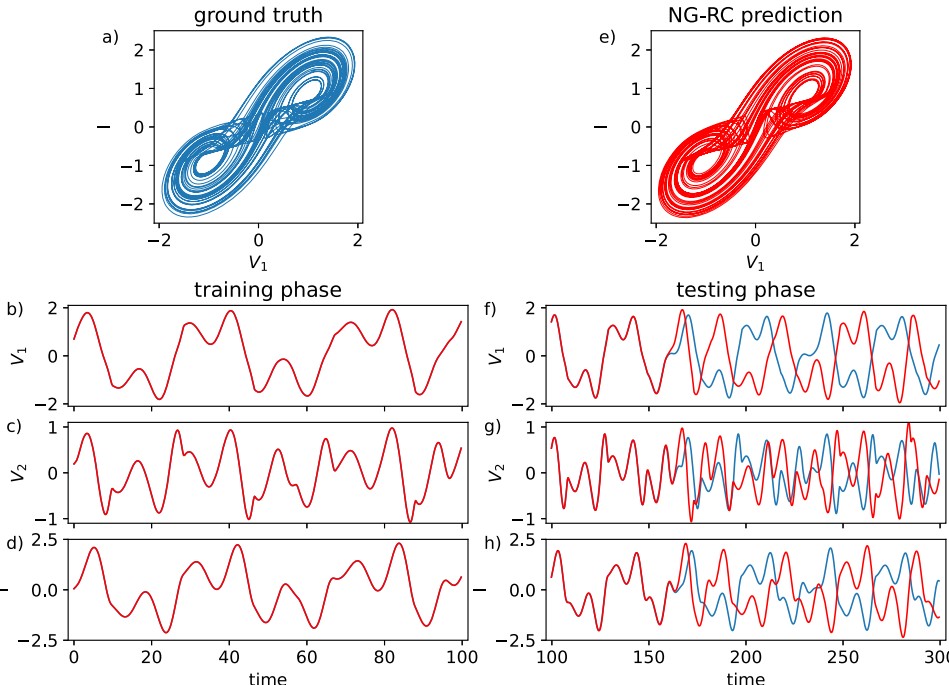

**Fig. 3 Forecasting the double-scroll system using the NG-RC.** True (**a**) and predicted (**e**) double-scroll strange attractors. **b–d** Training data set with overlayed predicted behavior. **f–h** True (blue) and predicted datasets during the forecasting phase (NRMSE $= 4.5 \pm 1.0 \times 10^{-3}$).

the Lorenz63 system. The results of this task are displayed in Fig. 3, where it is seen that the NG-RC shows similar predictive ability on the double-scroll system as in the Lorenz63 system, where other quantitative measures of accurate attractor reconstruction is given in Supplementary Note 1 as well as the components of $\mathbf{W}_{out}$ in Supplementary Note 2.

In the last task, we infer dynamics not seen by the NG-RC during the testing phase. Here, we use $k = 4$ and $s = 5$ with $dt = 0.05$ to generate an embedding of the full attractor to infer the other component, as informed by Takens' embedding theorem[29]. We provide the $x$, $y$, and $z$ variables during training and we again observe that a short training data set of only 400

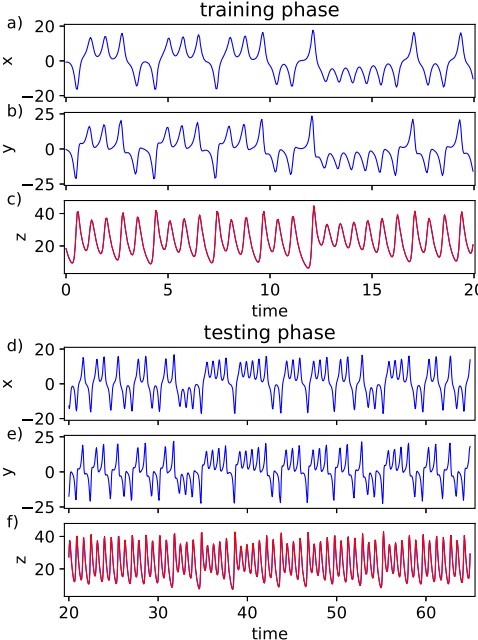

**Fig. 4 Inference using an NG-RC. a–c** Lorenz63 variables during the training phase (blue) and prediction (**c**, red). The predictions overlay the training data in (**c**), resulting in a purple trace (NRMSE = $9.5 \pm 0.1 \times 10^{-3}$ using $\alpha = 0.05$). **d–f** Lorenz63 variables during the testing phase, where the predictions overlay the training data in (**f**), resulting in a purple trace (NRMSE = $1.75 \pm 0.3 \times 10^{-2}$).

points is enough to obtain good performance as shown in Fig. 4c, where the training data set is overlayed with the NG-RC predictions. Here, the total feature vector has 45 components and hence $\mathbf{W}_{out}$ has dimension $(1 \times 45)$. During the testing phase, we only provide the NG-RC with the $x$ and $y$ components (Fig. 4d, e) and predict the $z$ component (Fig. 4f). The performance is nearly identical during the testing phase. The components of $\mathbf{W}_{out}$ for this task are given in Supplementary Note 2.

## Discussion

The NG-RC is computationally faster than a traditional RC because the feature vector size is much smaller, meaning there are fewer adjustable parameters that must be determined as discussed in Supplementary Notes 3 and 4. We believe that the training data set size is reduced precisely because there are fewer fit parameters. Also, as mentioned above, the warmup and training time is shorter, thus reducing the computational time. Finally, the NG-RC has fewer metaparameters to optimize, thus avoiding the computational costly optimization procedure in high-dimensional parameter space. As detailed in Supplementary Note 3, we estimate the computational complexity for the Lorenz63 forecasting task and find that the NG-RC is ~33–162 times less costly to simulate than a typical already efficient traditional RC[12], and over $10^6$ times less costly for a high-accuracy traditional RC[14] for a single set of metaparameters. For the double-scroll system, where the NG-RC has a cubic nonlinearity and hence more features, the improvement is a more modest factor of 8–41 than a typical efficient traditional RC[12] for a single set of metaparameters.

The NG-RC builds on previous work on nonlinear system identification. It is most closely related to multi-input, multiple-output nonlinear autoregression with exogenous inputs (NARX) studied since the 1980s[21]. A crucial distinction is that Tikhonov

regularization is not used in the NARX approach and there is no theoretical underpinning of a NARX to an implicit RC. Our NG-RC fuses the best of the NARX methods with modern regression methods, which is needed to obtain the good performance demonstrated here. We mention that Pyle et al.[30] recently found good performance with a simplified NG-RC but without the theoretical framework and justification presented here.

In other related work, there has been a revival of research on data-driven linearization methods[31] that represent the vector field by projecting onto a finite linear subspace spanned by simple functions, usually monomials. Notably, ref. 25 uses least-square while recent work uses LASSO[26,27] or information-theoretic methods[32] to simplify the model. The goal of these methods is to model the vector field from data, as opposed to the NG-RC developed here that forecasts over finite time steps and thus learns the flow of the dynamical system. In fact, some of the large-probability components of $\mathbf{W}_{out}$ (Supplementary Note 2) can be motivated by the terms in the vector field but many others are important, demonstrating that the NG-RC-learned flow is different from the vector field.

Some of the components of $\mathbf{W}_{out}$ are quite small, suggesting that several features can be removed using various methods without hurting the testing error. In the NARX literature[21], it is suggested that a practitioner start with the lowest number of terms in the feature vector and add terms one-by-one, keeping only those terms that reduce substantially the testing error based on an arbitrary cutoff in the observed error reduction. This procedure is tedious and ignores possible correlations in the components. Other theoretically justified approaches include using the LASSO or information-theoretic methods mentioned above. The other approach to reducing the size of the feature space is to use the kernel trick that is the core of ML via support vector machines[20]. This approach will only give a computational advantage when the dimension of $\mathbb{O}_{total}$ is much greater than the number of training data points, which is not the case in our studies here but may be relevant in other situations. We will explore these approaches in future research.

Our study only considers data generated by noise-free numerical simulations of models. It is precisely the use of regularized regression that makes this approach noise-tolerant: it identifies a model that is the best estimator of the underlying dynamics even with noise or uncertainty. We give results for forecasting the Lorenz63 system when it is strongly driven by noise in the Supplementary Note 5, where we observe that the NG-RC learns the equivalent noise-free system as long as $\alpha$ is increased demonstrating the importance of regularization.

We also only consider low-dimensional dynamical systems, but previous work forecasting complex high-dimensional spatial-temporal dynamics[4,7] using a traditional RC suggests that an NG-RC will excel at this task because of the implicit traditional RC but using smaller datasets and requiring optimizing fewer metaparameters. Furthermore, Pyle et al.[30] successfully forecast the behavior of a multi-scale spatial-temporal system using an approach similar to the NG-RC.

Our work has important implications for learning dynamical systems because there are fewer metaparameters to optimize and the NG-RC only requires extremely short datasets for training. Because the NG-RC has an underlying implicit (hidden) traditional RC, our results generalize to any system for which a standard RC has been applied previously. For example, the NG-RC can be used to create a digital twin for dynamical systems[33] using only observed data or by combining approximate models with observations for data assimilation[34,35]. It can also be used for nonlinear control of dynamical systems[36], which can be quickly adjusted to account for changes in the system, or for speeding up the simulation of turbulence[37].

## Methods

The exact numerical results presented here, such as unstable steady states (USSs) and NRMSE, will vary slightly depending on the precise software used to calculate them. We calculate the results for this paper using Python 3.7.9, NumPy 1.20.2, and SciPy 1.6.2 on an x86-64 CPU running Windows 10.

## Data availability

The data generated in this study can be recreated by running the publicly available code as described in the Code availability statement.

## Code availability

All code is available under an MIT License on Github (https://github.com/quantinfo/ng-rc-paper-code)[38].

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

## Acknowledgements

We gratefully acknowledge discussions with Henry Abarbanel, Ingo Fischer, and Kathy Lüdge. D.J.G. is supported by the United States Air Force AFRL/SBRK under Contract No. FA864921P0087. E.B. is supported by the ARO (N68164-EG) and DARPA.

## Author contributions

D.J.G. optimized the NG-RC, performed the simulations in the main text, and drafted the manuscript. E.B. conceptualized the connection between an RC and NVAR, helped interpret the data and edited the manuscript. A.G. and W.A.S.B. helped interpret the data and edited the manuscript.

## Competing interests

D.J.G. has financial interests as a cofounder of ResCon Technologies, LCC, which is commercializing RCs. The remaining authors declare no competing interests.
