## [Peer Review File · Nature Communications]

Reviewers' Comments:

Reviewer #1:

In their manuscript "Next Generation Reservoir Computing", Gauthier and co-authors propose a new neural network architecture with an application to dynamical and high-dimensional spatio-temporal systems in mind. They start by a comparison to reservoir computing (RC), which can be seen as a kind of benchmark neural network model for such dynamical system applications. The objective is to inject particular sub-sampled data of the target dynamical system and rely on the neural network in order to either predict the system's future evolution or some of its missing degrees of freedom. Other than in RC, the authors use an ad-hoc constructed network comprising concatenation of different polynomial orders of the input data. The input data combines the dynamical system's dimensions that are available from sampling, plus a number of their temporally shifted versions. As such, the network leverages principles of Takens attractor embedding. Training of the system is implemented using classical regression tools. The manuscript is well written and proposes an elegant alternative to current recurrent neural network approaches. I therefore find the proposed concept in principle of interest for Nature Communication. However, while the interpretability of their approach is of interest, the justification for high-impact publication heavily relies on generating significant benefits when compared to previous methods. At this level I must say that I remain unconvinced if this has been achieved.

The authors justify the relevance of their concept through (i) less required computational resources and (ii) faster training when compared to classical RC. RC in general is highly resource efficient, and the clear advantage of this approach relies on a potentially reduced of neural network output feature space O . At the moment the authors do not provide the data or the arguments to see as on which scale such a size reduction can be expected. For this one would, first, require a performance benchmarking with an average performance based on classical reservoirs. Second, the size of the output features space O scales with the required polynomial order, dimensionality and delay-dimensions. For example, the double scroll evaluation requires 3rd order polynomials and 3-4 system and delay dimensions. At third order polynomial scaling, this means that the feature size approaches potentially 100s of dimensions. How well would a reservoir with such a dimensionality do in comparison.

These kind of quantitative comparisons are essential in order to judge the relevance of the presented concept.

(ii) The authors claim more efficient training as they require a smaller training data-set. Is this a rather 'linear' improvement? How many samples would a classical RC require for similar accuracy? Also, could the authors please elaborate a bit more on their interpretation regarding the origin of the potentially reduced training samples size? In my understanding, and as stated by the authors themselves, for adequate training it is important to have a representative samples of the strange attractors phase space, independent of the prediction concept.

Smaller comments:

Fig. 1: the temporal shifts for the input sequence of the NG-RC are not correct, as they are $[t_{\{1+1\}}, t, t_{\{1+1\}}]$ instead of $[t_{\{1-1\}}, t, t_{\{1+1\}}]$.

"in a lest-square sense" -> "in a least-square sense"

"This is a dramatically shorter warm-up period in comparison to traditional RCs, where longer warm up times are needed to ensure that the reservoir state does not depends on the RC initial conditions." How justified is this statement in its generality? This will strongly depend on hyper parameters such as the system's leak-rate, etc.

Eq. (6). I am not sure I understand the meaning of the $\$p \text{ times} \$$ argument.

"explicitly or do any of the costly computation associated with it." I am not sure I can follow this claim. What do you mean with "any of the costly computation". Just as much as a reservoir states needs to be computed, you will need to compute your output feature space. Do you simply refer to that potential situation where you would require to initialize many random connection matrices? Please clarify.

"Task 3: Inferring unseeing Lorenz63 dynamics" -> "Task 3: Inferring unseen Lorenz63 dynamics"

Reviewer #2:

Remarks to the Author:

The manuscript is devoted to reservoir computing, an important and rapidly developing interdisciplinary topic that combines several fields, including machine learning and nonlinear complex systems. The main message of this work is very intriguing and has far-reaching implications: the nonlinear vector autoregression (NVAR), being equivalent to RC, outperforms the RC by using much smaller datasets for learning and fewer metaparameters. In particular, the authors show how to design a well-performing NVAR for three benchmark tasks usually addressed by RC.

In my opinion, the presented results can have a decisive impact on the research in the field of reservoir computing, and therefore, they are appropriate for Nature Communications. However, before making a definitive recommendation for publication, the authors should clarify the issues mentioned below.

In the Introduction:

1. The following statement:

"Recent work suggests that good matrices and metaparameters can be identified by determining whether the reservoir dynamics synchronizes in a generalized sense ..."

is misleading, because the generalized synchronization is one of the criteria, and it alone is not sufficient to determine good RC.

2. "... reproducing the long-term 'climate' of a chaotic system ..."

The notion of "climate" of a chaotic system is not commonly used in either ML or nonlinear dynamics. I suggest to rephrase the text passages where this term is used (or to introduce its meaning more clearly).

In Figure 1:

3. Ref (9) is not correct.

In the lower left inset, some of t_{i+1} should be t_{i-1} .

Otherwise, the Figure 1 is very informative and provides a very nice overview of the main principles of NG-RC and how it compares to RC.

In Section II:

4. Why are the mixed terms such as $r_1 * r_2$ not included in Eq (4)?

In Section III and Discussion:

4. Please discuss more about the choice of the meta-parameters k , s , dt . This choice looks very model-specific. Although the authors have provided some explanations for some cases, a general comment would be useful in the discussion.

5. The authors mentioned several times that the method learns the vector field. It was not clear to me in what sense the vector field was determined. To what extent does the prediction of the time-series implies the prediction of the vector field?

6. Robustness to noise is expected, but I would suggest that the authors at least provide a numerical evidence of such robustness. In my opinion, additional validation is useful, especially

given the high level of the journal.

7. L. 319 Patel -> Pyle

L. 335 Lüge -> Lüdge

8. In this paper, only low-dimensional chaotic systems are considered. Can the authors discuss the possibility of the prediction of high-dimensional chaos, which can be generated e.g. by a long-delayed feedback?

General formatting issue:

9. The formatting style of citations (x) is the same as that for the references to equations (x), and should therefore be changed.

Thank you for your detailed reading of our manuscript; your comments have greatly improved our work. Please find below detailed responses to your comments.

Reviewer #1 (Remarks to the Author):

In their manuscript “Next Generation Reservoir Computing”, Gauthier and co-authors propose a new neural network architecture with an application to dynamical and high-dimensional spatio-temporal systems in mind. They start by a comparison to reservoir computing (RC), which can be seen as a kind of benchmark neural network model for such dynamical system applications. The objective is to inject particular sub-sampled data of the target dynamical system and rely on the neural network in order to either predict the system’s future evolution or some of its missing degrees of freedom. Other than in RC, the authors use an ad-hoc constructed network comprising concatenation of different polynomial orders of the input data.

We agree that there is some *ad hoc* decision about truncating the polynomial order, but the polynomial expansion is a direct result of the discrete form of a convolution that serves as the universal representation of dynamical systems as discussed in Refs. 12 and 18. What is surprising in our work is that truncating the expansion at a low order works very well.

From the originally submitted manuscript, the following passage addresses this point:

“Polynomial functionals are the basis of a Volterra representation for dynamical systems (18) and hence they are a natural starting point. We find that low-order polynomials are enough to obtain high performance.”

The input data combines the dynamical system’s dimensions that are available from sampling, plus a number of their temporally shifted versions. As such, the network leverages principles of Takens attractor embedding. Training of the system is implemented using classical regression tools.

While we use time-delay versions of the data, we do not believe its relation to Takens’ embedding is the only way to understand the success of the NG-RC. For the two forecasting tasks we consider, we only take data from one time-step in the past ($s=1$, $k=2$ using our notation). We believe that a fruitful way to understand why this procedure is effective is considering methods of higher-order numerical integration methods. In these methods, the time interval is broken into sub-intervals and the final estimate of the integration step involves data from different past times over the interval. We do not sub-divide the interval as in these methods, but we think there is an analogy here.

For the inference task where we infer the z -variable of Lorenz63 from the x - and y -variables, we do agree that we are reconstructing the z -variable as a Takens-like embedding as we discuss in the text. Here, we “skip” several time intervals ($s > 1$) to include data further in the past and the chosen delay time is consistent with expectations if we were performing an embedding (such as optimizing the mutual information). In the original manuscript, we only mention Takens’ embedding theorem in the context of the inference task and not for the forecasting tasks.

We will expand on these thoughts and make them more rigorous in future publications.

The manuscript is well written and proposes an elegant alternative to current recurrent neural network approaches. I therefore find the proposed concept in principle of interest for Nature Communication.

We thank the referee for the support of our work.

However, while the interpretability of their approach is of interest, the justification for high-impact publication heavily relies on generating significant benefits when compared to previous methods. At this level I must say that I remain unconvinced if this has been achieved.

The authors justify the relevance of their concept through (i) less required computational resources and (ii) faster training when compared to classical RC.

RC in general is highly resource efficient, and the clear advantage of this approach relies on a potentially reduced of neural network output feature space O . At the moment the authors do not provide the data or the arguments to see as on which scale such a size reduction can be expected. For this one would, first, require a performance benchmarking with an average performance based on classical reservoirs.

For our tasks, there are already examples of applying a traditional RC to the forecasting or inference tasks, but most researchers do not give run time, which is highly dependent on the computational resources and computer language. To address the referee's comments, we derive the computational complexity of the two approaches in a new section of the Supplementary Materials, entitled *Comparing the computational complexity of the NG-RC with a typical traditional RC*. This analysis is less dependent on the particular hardware and software routines. We also added a paragraph to the beginning of the Discussion section:

“The NG-RC is computationally faster than a traditional RC because the feature vector size is much smaller, meaning there are fewer adjustable parameters that must be determined. We believe that the training data set size is reduced precisely because there are fewer fit parameters. Also, as mentioned above, the warmup and training time are shorter, thus reducing the computational time. Finally, the NG-RC has fewer metaparameters to optimize, thus avoiding the computationally costly optimization procedure in a high-dimensional parameter space. As detailed in the Supplemental Materials, we estimate the computational complexity for the Lorenz63 forecasting task and find that the NG-RC is approximately 33-162 times less costly to simulate than a typical efficient traditional RC⁹ and over 10^6 times less costly for a high-accuracy traditional RC¹⁰ for a single set of metaparameters. For the double-scroll system, where the NG-RC has a cubic nonlinearity and hence more features, the improvement is a more modest factor of 8-41 than a typical efficient traditional RC⁹ for a single set of metaparameters.”

Below, we repeat some of the comparison here to directly address the referee's comment.

Second, the size of the output features space O scales with the required polynomial order, dimensionality and delay-dimensions. For example, the double scroll evaluation requires 3rd order polynomials and 3-4 system and delay dimensions. At third order polynomial scaling, this means that the feature size approaches potentially 100s of dimensions. How well would a reservoir with such a dimensionality do in comparison.

In re-reading our manuscript, we noticed that we failed to give the values of k and s used for forecasting the double-scroll attractor. We are sorry for this omission; the following text is now added to Sec. V, Task 2: “and the use of the cubic monomials as given in Eq. 10 with $d=3$, $k=2$, and $s=1$ for a total of 62 features in \mathcal{O}_{total} ”. While we agree that the feature size grows combinatorial with the polynomial order, there are only 62 features in the total feature vector. This is much less than the 100 nodes in a traditional reservoir used by our group in Ref. 9 for the same double-scroll system, or the 300 features (4,000) used by Pathak *et al* in Ref. 3 (Lu *et al.* in Ref. 10).

Of course, going to higher-order polynomials will become much more expensive. However, we find that either a quadratic or a cubic is enough for all examples given in the manuscript and in several other systems we have studied recently but have not yet published. If low-order polynomials are not enough, the so-called kernel trick can be used to obtain high-order polynomial features (and even infinite order). We mention this point in the original manuscript in Sec. VI and discussed in Ref. 18.

But reducing the size of the feature vector is not the most important aspect of our computational savings. In a traditional reservoir computer, there are a multitude of metaparameters that need to be optimized as stressed in our original manuscript. For this optimization, the reservoir performance must be tested over and over again to find the best parameters. For example, for the double-scroll and Lorenz63 systems studied by our group in Ref. 9, there are 5 different metaparameters that need to be optimized (and others are held fixed to save computational time). Even using the Bayesian approach introduced in this work, there are many hours of computation time involved in this optimization. The NG-RC bypasses almost all of this work and it is easy to identify a high-performance reservoir computer.

These kind of quantitative comparisons are essential in order to judge the relevance of the presented concept.

Please see our comparison of the computational complexity in the new Supplementary Information section.

(ii) The authors claim more efficient training as they require a smaller training data-set. Is this a rather ‘linear’ improvement? How many samples would a classical RC require for similar accuracy?

This is a great question and we do not yet have an analytic estimate for the required data set size. This is an open research question that we are currently exploring. We give some scaling arguments here and then demonstrate they are roughly consistent with our observations.

From a dynamical systems perspective, you want to make sure that you sample enough of the attractor to obtain an accurate model of the flow of the system (the next-step-ahead mapping), which we mention in the original manuscript. But we know of no argument for how many points are enough.

Another window into the data requirements is that we need to specify the number of unknowns that must be fit in the linear regression.

For the Loren63 forecasting task (Task 1 in the manuscript), we have 28 components of the feature vector and we are predicting 3 variables for a total of 84 unknowns that need to be determined. For the double-scroll forecasting task, we have 62 components of the feature vector and 3 variables for a total 186 unknowns. The minimum training data then should likely be at least this size. However, to have the network generalize, somewhat more data is probably needed.

For the Lorenz63 forecasting task, we see the testing error drop approximately linearly with training data set size until we get to about 400 data points – what we used in the manuscript – and then there is much slower reduction of error with additional data. These points are mentioned in the original Supplementary Materials in the section on *Return map* above Fig. 5.

With regards to the training data required for a traditional reservoir computer, we are not aware of a systematic study. In our own work in Ref. 9, we use 10,000 data points, Pathak *et al.* (Ref. 3) uses 5,000, and Lu *et al.* (Ref. 10) uses 60,000. In our own work, we certainly adjusted the training time, and our choice was a tradeoff between accuracy and computation time. We have not published any results on

performance vs. training time, but we often see that the error scales inversely with the square root of the number of data points. We see rather different behavior with the NG-RC as stated above, which we speculate is due to near optimality of the network for the NG-RC.

We hesitate to say more in the manuscript as we do in the previous paragraphs because we do not have a solid theory to predict the required training data set size and we do not want to put speculation in the manuscript. But we hope that the referee is satisfied with our scaling arguments.

Also, could the authors please elaborate a bit more on their interpretation regarding the origin of the potentially reduced training samples size? In my understanding, and as stated by the authors themselves, for adequate training it is important to have a representative samples of the strange attractors phase space, independent of the prediction concept.

Please see discussion above for the previous comment.

Smaller comments:

Fig. 1: the temporal shifts for the input sequence of the NG-RC are not correct, as they are $[t_{\{1+1\}}, t_{\{1+1\}}]$ instead of $[t_{\{1-1\}}, t_{\{1+1\}}]$.

Thank you for pointing out this error. The figure is fixed.

“in a lest-square sense” -> “in a least-square sense”

Fixed.

“This is a dramatically shorter warm-up period in comparison to traditional RCs, where longer warm up times are needed to ensure that the reservoir state does not depends on the RC initial conditions.” How justified is this statement in its generality? This will strongly depend on hyper parameters such as the system’s leak-rate, etc.

We agree that it depends on the traditional RC metaparameters. For an interconnected reservoir, the dynamics are most strongly dependent on the spectral radius of the connectivity matrix, not just the leak rate. That is, their will almost certainly be a Floquet multiplier (the appropriate stability index for a mapping such as Eq. 1) with a magnitude near 1 when the spectral radius is set near 1, which is often needed to obtain good performance.

To make a comparison to some of the literature, we have added the following sentence to the manuscript right after we discuss the warm-up time (the paragraph after Eq. 5).

“For example, with $s=1$ and $k=2$ as used for some examples below, only 2 warm up data points are needed. A typical warm up time in traditional RC for the same task can be upwards of 10^4 to 10^5 data points^{3,9,10}.”

Eq. (6). I am not sure I understand the meaning of the \mathbb{P} times argument.

We use a subscript (p) for the nonlinear feature vector as $\mathbb{O}_{nonlinear}^{(p)}$ in analogy of the (2) for the quadratic polynomial whose expression is given explicitly. The ellipses in Eq. 6 means that Olin appears “ p times” and the operator $[\otimes]$ is applied $(p-1)$ times. To clarify the equation, we have

change the text after the equation reads: “*with \mathcal{O}_{lin} appearing p times.*” We believe this is now more precise.

“explicitly or do any of the costly computation associated with it.” I am not sure I can follow this claim. What do you mean with “any of the costly computation”. Just as much as a reservoir states needs to be computed, you will need to compute your output feature space. Do you simply refer to that potential situation where you would require to initialize many random connection matrices? Please clarify.

Some of this has been address above already and we briefly summarize here. In our own work on optimizing traditional RCs (Ref. 9), we optimized 5 additional metaparameters that are part of the traditional RC but not the NG-RC: the node leak rate, the range of random numbers of the input connectivity matrix, the probability that an input node is connected to a reservoir node, the spectral radius, and the connectivity of the reservoir (the sparsity of the connectivity matrix). This is a very high-dimensional search space; each reservoir needs to be trained and its performance tested for each point in this space. The whole point of Ref. 9 is to use Bayesian optimization to speed up this process, but it still requires many minutes to hours of computation time. On top of this, we also have to optimize the ridge parameter α , which we do in Ref. 9 through cross-validation; even this step is computationally expensive.

Even with optimizing the parameters, there is still a distribution of performances depending on the specific topology defined by the random matrices. In studying 100 different optimized realizations, we always find some outliers that perform $>10x$ worse that the median performer due only to the different random topologies.

Our statement in the manuscript and which is quote above in the comment reflects the fact that most of these parameters are not present in the NG-RC, and there are no random matrices and hence we always find the best performer.

We agree that we still need to train and test and optimize the ridge parameter, but that is the only parameter out of the many others that are present in a traditional RC. We believe that our statement is accurate as stated.

“Task 3: Inferring unseeing Lorenz63 dynamics” -> “Task 3: Inferring unseen Lorenz63 dynamics”

Fixed.

Reviewer #2 (Remarks to the Author):

The manuscript is devoted to reservoir computing, an important and rapidly developing interdisciplinary topic that combines several fields, including machine learning and nonlinear complex systems. The main message of this work is very intriguing and has far-reaching implications: the nonlinear vector autoregression (NVAR), being equivalent to RC, outperforms the RC by using much smaller datasets for learning and fewer metaparameters. In particular, the authors show how to design a well-performing NVAR

for three benchmark tasks usually addressed by RC.

In my opinion, the presented results can have a decisive impact on the research in the field of reservoir computing, and therefore, they are appropriate for Nature Communications.

We thank the referee for support of our work and we are glad that our text properly highlights these main points, which we also consider to be our take-home messages.

However, before making a definitive recommendation for publication, the authors should clarify the issues mentioned below.

In the Introduction:

1. The following statement:

“Recent work suggests that good matrices and metaparameters can be identified by determining whether the reservoir dynamics synchronizes in a generalized sense ...”

is misleading, because the generalized synchronization is one of the criteria, and it alone is not sufficient to determine good RC.

This sentence is meant to describe our understanding of Ref. 10. In the discussion after Eq. 7 of Ref. 10, they make strong statements that obtaining generalized synchronization is enough to guarantee that a reservoir computer can succeed in the task of attractor reconstruction, implying that the other criteria mentioned by the referee is not needed. But the authors of Ref. 10 concede that generalized synchronization may be too strong of a requirement, but they do claim that a reservoir that does not display generalized synchronization will fail to reconstruct (forecast) an attractor.

But our purpose is not to argue a point that is not even relevant to the NG-RC. We are happy to take a suggestion from the referee on how to change this sentence, but we think our use of the word “suggests” weakens the statement considerably. We are not saying that this is a definitive proof or a necessary and sufficient condition, but that there is an interesting or “suggestive” line of recent research pointing to a possible future comprehensive theory.

2. “... reproducing the long-term ‘climate’ of a chaotic system ...”

The notion of "climate" of a chaotic system is not commonly used in either ML or nonlinear dynamics. I suggest to rephrase the text passages where this term is used (or to introduce its meaning more clearly).

This phraseology is already in the reservoir computing literature, first found in Ref. 10 and appearing in several more recent works from the U Maryland group and by people citing Ref. 10. We believe this is an established term.

But to make our work more accessible, we have added the text to this sentence in the introduction:

“ ... reproducing the long-term ‘climate’ of a chaotic system (**that is, reconstructing** the attractors shown in Fig. 1);”

This follows the definition used in Ref. 10.

In Figure 1:

3. Ref (9) is not correct.

In the lower left inset, some of t_{i+1} should be t_{i-1} .

Fixed. Also, we used the incorrect citation in the figure and this is also fixed.

Otherwise, the Figure 1 is very informative and provides a very nice overview of the main principles of NG-RC and how it compares to RC.

Thank you.

In Section II:

4. Why are the mixed terms such as $r_1 * r_2$ not included in Eq (4)?

We are just reporting what has been done in the past literature, where only a Hadamard product is used. See Refs. 3, 9, 10, etc. The motivation in this work is that they want to break the symmetry of the output feature vector and the Hadamard operation is enough to do this. Perhaps more interesting is that the Hadamard product in the traditional RC transforms to the unique terms of the outer product in the NG-RC as shown by co-author Bollt in Ref. 14. For the NG-RC, the terms in the output feature vector need to represent terms in a Volterra series (or convolutions) and hence you need all these cross terms. Stated another way, the NG-RC with all the cross terms is mathematically equivalent to traditional RC using the Hadamard product. So perhaps we are the first to provide insight into why the past works found heuristically that the Hadamard product gives high-performing RCs given that the two approaches are mathematically equivalent.

In Section III and Discussion:

4. Please discuss more about the choice of the meta-parameters k , s , dt . This choice looks very model-specific. Although the authors have provided some explanations for some cases, a general comment would be useful in the discussion.

For the forecasting task, we find $s=1$ works well assuming that dt is chosen appropriately and this is not sensitive to the specific model. With regards to selecting dt , the data is highly correlated when dt is too small and hence this is not much gain in performance relative to the increase computational cost for processing more data. We have not carefully studied how the

largeness of dt affects the NG-RC performance; we found good performance for larger values of dt than we used in the manuscript, but then the figures of the attractor looked “jumpy” (and this is true for the ground-truth attractor). We selected the largeness of dt merely to make a good figure, not because the performance degrades. I suppose we could have used an interpolating function to smooth the plots, but we did not do this. In summary, these parameters are not all that sensitive and do not appear to be model dependent.

The issue related to how far we take data in the past (parameter k) is more interesting. Based on the existing theory, we should take k toward infinity. But practically, many people have noticed that the Volterra representation of functions converges quickly. This is likely also the origin of why numerical integration schemes for ordinary differential equations only requires a few sub-intervals to obtain high accuracy. To make these points, we have added the following paragraph after Eq. 5.

Based on the general theory of universal approximators^{12,16}, k should be taken infinitely large. However, it is found in practice that the Volterra series converges rapidly and hence truncating k small does not incur large error. This can also be motivated by considering numerical integration methods of ordinary differential equations where only a few sub-intervals (steps) in a multi-step integrator are needed to obtain high accuracy. We do not sub-divide the step here, but this analogy motivates why small values of k might give good performance in the forecasting tasks considered below.

5. The authors mentioned several times that the method learns the vector field. It was not clear to me in what sense the vector field was determined. To what extent does the prediction of the time-series implies the prediction of the vector field?

Actually, we state that the NG-RC does NOT learn the vector field (the right-hand-side of a set of differential equations). What we state is that the NG-RC learns one-step-ahead prediction. Formally, a set of differential equations can be written in terms of a map that takes the system from one point in phase space to another point in phase space at a future time. This is called the flow of the dynamical system. The flow involves an exponential of an integral of the vector field. To clarify this point, we have modified the first few sentences after Eq. (10) to read:

For these forecasting tasks, the NG-RC learns simultaneously the vector field and an efficient one-step-ahead integrator to find a mapping from one time to the next without having to learn each separately as in other nonlinear state estimation approaches^{21–24}. The one-step-ahead mapping is known as the *flow* of the dynamical system and hence the NG-RC learns the flow. We also add the following sentence at the end of the results for Task 1 when discussing the components of \mathbf{W}_{out} :

“This gives quantitative information about the difference between the flow and the vector field.”

6. Robustness to noise is expected, but I would suggest that the authors at least provide a numerical evidence of such robustness. In my opinion, additional validation is useful, especially given the high level of the journal.

See new section in the Supplementary Materials, entitled *NG-RC performance for the Lorenz63 system driven by noise*. Also see the sentence added to the discussion section:

“We give results for forecasting the Lorenz63 system when it is strongly driven by noise in the Supplementary Materials, where we observe that the NG-RC learns the equivalent noise-free system as long as α is increased demonstrating the importance of regularization.”

Here, we add terms to the Lorenz63 differential equations that represent Gaussian-distributed noise for each variable. The rms value of the noise is 1, which should be compared to the rms values for each variable in the absence of noise, which is around 8.5. Thus, the noise terms are very large – about 12% of the typical size of the variables.

We train on the noise time series and then compare the predicted dynamics from the NG-RC to the *noise-free* evolution of Lorenz63. We obtain an NRMSE of 1.4% over the first Lyapunov time obtained by increasing the ridge parameter to prevent overfitting to the noisy data. Thus, the NG-RC learns the underlying dynamical model and is a way to have a “noise free” prediction of the dynamics. The ability of machine learning routines to perform this type of task has been discussed extensively by Billing’s group (see Ref. 17) in the context of the NARX. However, they need an explicit estimate of the noise, which is included in the NARX feature vector. We did not do this in the NG-RC formulation; we merely increased the ridge parameter.

7. L. 319 Patel -> Pyle

L. 335 Lüge -> Lüdge

Thank you for catching these. Both are fixed.

8. In this paper, only low-dimensional chaotic systems are considered. Can the authors discuss the possibility of the prediction of high-dimensional chaos, which can be generated e.g. by a long-delayed feedback?

We are saving some of this work for future publication, but we find that it is straightforward to predict the dynamics of time-delay systems, such as the Mackey-Glass time-delay differential equation, using only a small feature vector.

Also, the Pyle et al. work shows a truncated version of the NG-RC can be used to predict 1-dimensional spatial-temporal systems, such as the Lorenz96 system with nearly 100 nodes. We mentioned this work in the discussion section in the original manuscript.

Finally, we remark that the U Maryland group devised an ingenious method for scaling RCs to forecast spatial-temporal dynamics using small parallel traditional RCs that cover a small spatial region with some overlap with its neighbors. See our Ref. 4. Given that the NG-RC is mathematically equivalent to a traditional RC but more compact and optimized, we should be able to play the same trick but with vastly reduced training data and computational time. We are currently working on this problem and will report the results in a future publication.

General formatting issue:

9. The formatting style of citations (x) is the same as that for the references to equations (x), and should therefore be changed.

The revised manuscript now uses the Nature reference format, which fixes this problem.

Reviewers' Comments:

Reviewer #1:

Remarks to the Author:

I would like to thank the authors for the clear and detail reply to my previous comments. All my suggestions have been addressed in a satisfactory manner.

I have one final comment: the authors claim that "Even using the Bayesian approach introduced in this work", which is not correct. Bayesian optimization of hyper-parameters was introduced by Antonik et al., "Bayesian optimisation of large-scale photonic reservoir computers," Cognitive Computation (2021) - which is available on arxiv since end 2019. I would like to ask the authors to cite this article at the relevant section.

Otherwise I am happy to recommend publication.

Reviewer #2:

Remarks to the Author:

I am satisfied by the feedback to my comments and the inclusion of new results. I recommend publication of the manuscript as is.

Thank you for your detailed second reading of our manuscript. Please find below detailed responses to your comments.

Reviewer #1 (Remarks to the Author):

I would like to thank the authors for the clear and detail reply to my previous comments. All my suggestions have been addressed in a satisfactory manner.

I have one final comment: the authors claim that "Even using the Bayesian approach introduced in this work", which is not correct. Bayesian optimization of hyper-parameters was introduced by Antonik et al., "Bayesian optimisation of large-scale photonic reservoir computers," Cognitive Computation (2021) - which is available on arxiv since end 2019. I would like to ask the authors to cite this article at the relevant section.

Otherwise I am happy to recommend publication.

Author Response:

We thank the referee for positive comments on our changes to the manuscript.

The referee provides a quote above. That statement does not appear in the main text or supplementary documents but does appear in our first response to the referees. Thus, no change is needed to the manuscript or the supplementary information, but we do agree that our response to the referees misstated the priority of the current Ref. 12 of using Bayesian optimization for reservoir computers.

In fact, the Antonik paper is not the first to use a Bayesian approach for optimizing a reservoir computer as stated by the referee. Both the current Ref. 12 and the Antonik paper cite a work appearing only on the arXiv (as far as we are aware) by Yperman & Becker from 2016. There are also optimization approaches using gradient descent and Fisher information. We have added citations 9 (Yperman & Becker, Bayesian optimization), 10 (Livi *et al.*, Fisher information optimization), 11 (Thiede & Parlitz, gradient descent optimization), and 12 (Antonik et al., Bayesian optimization).

Reviewer #2 (Remarks to the Author):

I am satisfied by the feedback to my comments and the inclusion of new results. I recommend publication of the manuscript as is.

Author Response:

We thank the referee for positive comments on our changes to the manuscript.

No changes to the manuscript are required.